# Parental Effect on Agronomic and Olive Oil Traits in Olive Progenies from Reciprocal Crosses

**DOI:** 10.3390/plants13172467

**Published:** 2024-09-03

**Authors:** Hristofor Miho, Mihad Atallah, Carlos Trapero, Georgios Koubouris, Pedro Valverde

**Affiliations:** 1Agronomy Department, University of Cordoba (UCO), 14005 Cordoba, Spain; hmiho@uco.es (H.M.); z12atatm@uco.com (M.A.); g32trrac@uco.es (C.T.); 2Santa Cruz Ingeniería S.L., 41018 Sevilla, Spain; 3Hellenic Agricultural Organization ELGO-DIMITRA, Institute for Olive Tree Subtropical Crops and Viticulture, 73134 Chania, Greece; koubouris@elgo.gr; 4Department of Agricultural, Food and Environmental Sciences, Marche Polytechnic University, 60121 Ancona, Italy

**Keywords:** breeding, crossing, maternal effect, fatty acids, fruit, phenols, *Olea europaea*, stability, sterols, virgin olive oil

## Abstract

Olive growing is undergoing a transition from traditional cultivation systems to a more technological model characterized by increased mechanization and a higher density of plants per hectare. This shift implies the use of less vigorous varieties that can adapt to the new system. Most traditional varieties are highly vigorous, and breeding programs can provide solutions to this challenge. This study investigates the parental effect on different agronomic and olive oil characteristics and its role in breeding programs. The objectives were to evaluate and characterize different agronomic and olive oil traits in the progenies from ‘Arbosana’ × ‘Sikitita’ cross and its reciprocal cross ‘Sikitita’ × ‘Arbosana’. The results showed a high variability of the characters evaluated in the progenitors of the reciprocal crosses. The highest coefficients of variation were observed in traits related to ripening index, phenolic compounds, polyunsaturated fatty acids, and Δ5-avenasterol, with phenolic content exhibiting the greatest variability. No statistically significant maternal effect was detected for any of the evaluated traits, although a slight positive maternal effect was systematically observed in the mean values of the evaluated traits. These results suggest that the maternal effect on olive is quite subtle, although due to a slight tendency of the maternal effect in the descriptive analyses, future studies are suggested to understand in depth the possible maternal effect on olive breeding.

## 1. Introduction

Olive cultivation has a rich history spanning centuries, with farmers primarily in the Mediterranean Basin meticulously selecting and developing various olive cultivars to enhance olive production and oil yield. This lengthy process has also aimed to adapt olive trees to diverse edaphoclimatic conditions across different growing areas [1]. Such extensive selection has resulted in a wide variety of vigorous olive cultivars, each with unique elaiotechnical characteristics, contributing significantly to the agricultural and economic landscape of the Mediterranean region [1,2].

The olive tree (*Olea europaea* L.) is a crucial crop in Mediterranean agriculture, with significant cultural, economic, and nutritional value. In recent years, the olive industry has evolved from traditional, low-mechanization practices to super-intensive, highly mechanized systems. This evolution requires new olive cultivars with trees that are small enough for machine harvesting, while also ensuring high yield, nutritional oil, and resistance to various biotic and abiotic stresses [3,4,5].

Virgin olive oil (VOO), extracted freshly from olives, preserves the natural nutritional properties of the fruit [2,6]. The quality of this oil is determined by a complex combination of chemical, physical, and sensory attributes, which allow its classification into various categories as defined by the International Olive Oil Council (IOC). The stability and nutritional value of olive oil are significantly influenced by its composition, especially the minor compounds such as phenols [7,8]. Renowned for its health benefits, olive oil is rich in monounsaturated fatty acids and phenolic compounds that provide antioxidant and anti-inflammatory properties [5,9].

Previous studies have shown different heritability rates depending on the evaluated trait; for example, a higher heritability has been observed in the fatty acid profile compared to the phenolic content, since phenols are more affected by environmental factors [10,11,12].

Despite the economic importance of olives, the genetic inheritance of key agronomic traits remains poorly understood. To advance breeding programs, it is essential to understand the genetic and environmental factors influencing these traits. Key objectives in olive breeding include early fruiting, maximizing yield, and adapting to high-density planting methods suitable for mechanical harvesting [13]. Modern breeding programs have also focused on disease resistance, such as resistance to *Verticillium* wilt [3,14]. The demand for new cultivars suitable for modern cultivation methods and changing climatic conditions necessitates genetic studies and precision breeding that decompose phenotypic variance into its genetic and environmental components [15,16,17]. Also, plant breeding is speeding up due to the power of genetic and molecular tools, such as the case of rosaceous crops, where DNA-informed strategies like marker-assisted parent selection (MAPS) and marker-assisted seedling selection (MASS) have improved breeding efficiency [18,19].

Heritability in plant breeding quantifies the proportion of trait variability attributed to genetic factors, affecting the similarity among relatives [20,21]. The success of breeding programs depends on selecting appropriate progenitors and understanding the inheritance patterns of target traits [2,22].

The maternal effect in genetics refers to the influence of the maternal parent on the offspring’s phenotype, often deviating from the expected equal chromosomal contribution from each parent [23,24,25]. Reciprocal crosses in plants can reveal phenotypic differences between hybrids of the same generation, such as F1 or F2, due to variations in cytoplasmic genomes between parents. Additionally, studies like Henry Hadley’s 1974 research have demonstrated maternal effects on protein content in hybrid seeds. These effects stem from both cytoplasmic and nuclear influences, which were observed through differences in reciprocal crosses across multiple generations, indicating a complex interplay of maternal factors in determining phenotypic traits [24,25,26].

Emphasizing the significant role of maternal factors, studies across various crops have investigated the influence of reciprocal crosses. For example, in peaches, Dini et al. (2021) [27] observed differences in phenological trait heritability based on the female or male parent, suggesting a potential maternal effect. Similarly, research on cacao and apple breeding has highlighted notable reciprocal variations and maternal effects [28,29,30].

The hereditary behavior of *Olea europaea* has been studied in olive genotypes from several crosses. Some works studied the parental effect in free pollination crosses focusing on the maternal effect and determining dominant female heredity for certain vegetative characteristics, including tree size and leaf and fruit shape [31]. Recent studies using reciprocal crosses have determined that there is no material effect on resistance to *Verticillium* wilt in olive progenies [14] and other characteristics such as olive oil content, first flowering, tree vigor, and tree size [32]. Another study found that genotypes resulting from reciprocal crosses between ‘Arbequina’ and ‘Picual’ showed predominantly intermediate values for flower quality traits, with no notable distinctions between maternal and paternal influences [33]. León et al. (2005) observed broad variation in fruit removal force among offspring from reciprocal crosses involving ‘Arbequina’, ‘Frantoio’, and ‘Picual’, with no significant distinctions among the female parent varieties [34].

Olive breeding helps conserve and improve specific characteristics of cultivars that demonstrate adaptability to environmental conditions or possess valuable intrinsic qualities. Understanding the genetic basis underlying the variability of agronomic traits in cultivated olives is essential for selecting new varieties that meet evolving societal needs and changing environmental conditions [35].

Recently, the hedgerow olive system has gained recognition as a viable method for olive cultivation. However, most traditional cultivars are not well suited for high-density hedgerow cultivation [36,37]. Despite this, some cultivars like ‘Arbequina’ and ‘Arbosana’ are extensively used in the super high-density (SHD) system [38]. Newer varieties such as ‘Sikitita’ [39], ‘Lecciana’ [40], and ‘Sultana’ [41] have been developed specifically for this cultivation system through breeding programs. These cultivars exhibit lower vigor, earlier bearing, and reduced alternate bearing tendencies compared to other varieties [42,43,44,45,46,47,48,49]. However, the adaptation of new varieties to this system remains a priority to increase the options available to farmers. Knowledge of the inheritance of traits from parents to seedlings is scarce, particularly regarding olive oil quality characteristics.

To address the knowledge gap concerning the inheritance of traits related to olive agronomic traits and especially oil quality, this study aims to elucidate how specific characteristics are inherited from parent plants to seedlings. The objectives of the study were to investigate the influence of female and male parents on the inheritance of key agronomic traits, including tree vigor, fruit weight, stone weight, and fat yield, as well as crucial chemical characteristics of olive oil, such as the phenolic profile, stability, fatty acid composition, and sterol content. To achieve that, we conducted a comprehensive evaluation of 32 genotypes derived from reciprocal crosses between the olive cultivars ‘Sikitita’ and ‘Arbosana’ by comparing the seedling of the ‘Sikitita’ × ‘Arbosana’ and ‘Arbosana’ × ‘Sikitita’ crosses. This research seeks to provide valuable insights into the inheritance patterns of these traits, thereby contributing to the optimization of olive breeding programs and cultivation practices.

## 2. Materials and Methods

### 2.1. Plant Material and Experimental Plot

In February 2015, a comprehensive experimental trial located in Almensilla, Sevilla, Spain (coordinates: 37°18′51.0″ N; 6°07′28.7″ W), was established utilizing a randomized block design. The planting frame used was 4 m between rows and 1.5 m between plants. The climate is Mediterranean with oceanic influences with an average rainfall of 493.4 mm, a location at 58.1 m above sea level, an average maximum temperature of 24.5 °C, an average medium temperature of 17.5 °C, and average minimum temperature of 11.5 °C, and a soil structure of loam-clay. The field have a deep irrigation system with a water dose of 1500 m^3^/ha/year. The plantation was carried out with one-year-old plants with a height of 60 cm and a trunk diameter of around 4 mm.

This trial aimed to evaluate several agronomic characteristics in the reference well-adapted ‘Arbequina’ and ‘Arbosana’ varieties, known for their suitability to hedgerow cultivation [41], alongside 32 different genotypes. These genotypes comprised 16 from the ‘Arbosana’ × ‘Sikitita’ crossing and 16 from the reciprocal ‘Sikitita’ × ‘Arbosana’ crossing, obtained by applying pollen to previously bagged branches with flowers according to [3]. Paternity tests by SSR DNA markers were carried out to validate the identity of the genitors in the selected genotypes [50]. The experimental design consisted of 4 random blocks, each containing 1 plant from each control cultivar and 4 genotypes from each cross, summing up to 10 plants per block and 40 plants for the entire trial.

### 2.2. Assessment of Agronomic Characteristics

The agronomic characteristics assessed related to the tree were plant vigor, in terms of height and width (measured in meters with a telescopic metric rod), and trunk diameter (in millimeters, measured with an electronic caliber). For the characterization of the fruits, 50 fruits of each plant were collected and transported in refrigerators to the laboratory for testing. Fruit weight and stone weight (in grams) were weighing using an electronic balance. The stone weight was determined by manually removing all the pulp and weighing the fresh stone immediately. The pulp-to-stone ratio was calculated as the percentage of fruit pulp weight relative to the total fruit weight.

To determine the fruit olive oil content (yield), a sample of 500 g per plant was collected and transported to the laboratory. The total percentage of olive oil in both fresh and dry weight, as well as the percentage of humidity, was determined using MiniSpec MQ-10, Bruker, Madison, WI, USA, functioning with nuclear magnetic resonance [51]. Additionally, the average fruit maturity index for each genotype was assessed on a scale from 0 to 4, where 0 = green fruit; 1 = yellow fruit; 2 = veraison fruit; 3 = purple fruit; and 4 = black fruit [4]. Vigor measurements and fruit sampling were conducted in November 2023.

### 2.3. Oil Quality Parameter Characterization

#### 2.3.1. Olive Oil Extraction

For determining oil chemical characteristics, in November 2023, samples of 2 kg of fruit per tree were collected through homogeneous sampling around the entire tree crown.

Virgin olive oil (VOO) samples were extracted using Abencor equipment under optimized conditions: 1 kg of olives per sample was milled, with a malaxation time of 30 min at 28 °C, followed by centrifugation at 3500 rpm for 90 s [52]. No water was added throughout the process to minimize the loss of phenolic compounds.

#### 2.3.2. Analysis of Phenolic Profiles

##### Reagents and Standards

For the analysis and quantification of phenolic compounds in virgin olive oils (VOOs), we employed MS-grade methanol (MeOH), n-hexane, and formic acid, all sourced from Scharlab (Barcelona, Spain). Deionized water (18 MΩ· cm) was prepared using a Millipore Milli-Q water purification system (Millipore, Bedford, MA, USA) and was utilized to create both aqueous and organic mobile phases. A hydroalcoholic mixture was used as the extractant for the samples. The phenolic compounds quantified included hydroxytyrosol, oleacein (3,4-DHPEA-EDA), oleocanthal (p-HPEA-EDA), oleuropein aglycone (3,4-DHPEA-EA), ligstroside aglycone (p-HPEA-EA), luteolin, and apigenin. Standards for hydroxytyrosol, apigenin, and luteolin were obtained from Extrasynthese (Genay, France), while standards for oleacein, oleocanthal, and the monoaldehyde forms of oleuropein and ligstroside aglycones were provided by Professor Prokopios Magiatis (University of Athens, Greece). The open aldehyde forms were quantified using the corresponding closed monoaldehyde standards. Standard solutions were prepared at 1 mg/mL in methanol for non-secoiridoid phenols and in pure acetonitrile for secoiridoids to maintain their stability.

##### Sample Preparation

Phenolic compounds were extracted from VOOs using a liquid–liquid extraction method as described by Sánchez de Medina et al. (2015) [53]. One gram of VOO was combined with 2 mL of n-hexane, followed by the addition of 1 mL of a 60:40 (*v*/*v*) methanol–water mixture. The mixture was shaken for 2 min, and the hydroalcoholic phase was separated via centrifugation. To enhance extraction efficiency, the process was repeated, and the combined extracts were analyzed. Syringaldehyde was used as an internal standard at a concentration of 2 mg/L. The extracts were then subjected to LC–QqQ-MS/MS analysis, with dilution factors of 1:2 and 1:50 (*v*/*v*) to accommodate a broad concentration range.

##### LC–MS/MS Analysis

The LC–MS/MS analysis was conducted using an Agilent 6475 Triple Quadrupole LC/MS system. Reversed-phase liquid chromatography was performed with a C18 Pursuit XRs Ultra column (50 mm × 2.0 mm i.d., 2.8 µm particle size) from Varian (Walnut Creek, CA, USA). The column temperature was maintained at 30 °C. The mobile phases consisted of 0.1% (*v*/*v*) formic acid in water (phase A) and 0.1% (*v*/*v*) formic acid in MeOH (phase B). The gradient program operated at a flow rate of 0.4 mL/min with the following steps: initial conditions of 50% phase A and 50% phase B for 0.5 min, followed by a decrease in phase A from 50% to 20% over 2 min, and further reduction to 0% over the next 2 min, with this final composition held for 1 min. The column was re-equilibrated to the initial conditions for 5 min. The eluate was ionized using electrospray ionization (Santa Clara, CA, USA) in negative mode and monitored using MS/MS in multiple reaction monitoring (MRM) mode, with specific parameters for each analyte detailed in Appendix A. Nitrogen was used as the drying gas at a flow rate of 10 L/min and a temperature of 300 °C, with a nebulizer pressure of 50 psi and a capillary voltage of 3000 V.

Calibration curves were generated using refined sunflower oil spiked with phenolic standards at nine concentration levels, ranging from 0.1 μg/mL to 10 μg/mL. Each concentration was analyzed in triplicate, including the entire sample preparation process. The refined oil was confirmed to be free of phenols through direct analysis. The calibration equations were then used to calculate the absolute concentrations of target phenols in the VOO samples. The “Sum of phenols” parameter was determined by summing the individual concentrations of the detected phenols. Each VOO sample was analyzed in triplicate to obtain the mean concentration.

#### 2.3.3. Determination of the Oxidation Stability Index in VOO Samples

The oxidation stability index of the VOO samples was measured using the Rancimat Method, employing a 743 Rancimat System from Metrohm (Herisau, Switzerland). In this analysis, 3.2 g of each VOO sample was heated to 100 °C with an air flow rate of 10 L/h to determine the time (in hours) until oil oxidation occurred. This analysis was carried out at the Agricultural and Food Laboratory of Cordoba (Cordoba, Spain), the designated reference laboratory for the Andalusia Regional Government.

#### 2.3.4. Analysis of Fatty Acid Profiles

The fatty acid profile was obtained using a gas chromatograph coupled to a flame ionization detector (GC-FID). A 0.05 g aliquot of each VOO sample was vortexed with 250 μL of 0.5 N methanolic potassium hydroxide at 30 °C for 10 min at 1700 rpm for transesterification. Next, 500 μL of n-hexane was added and stirred for 5 min to obtain a biphasic system. A 50 μL aliquot of the hexane phase containing the fatty acid methyl esters (FAMES) was collected and diluted 1/6 (*v*/*v*) before injecting 1 μL into the GC-FID. FAMES separation was carried out with a 7820A GC (Agilent Technologies, Santa Clara, CA, USA) equipped with an autosampler and a split/splitless injector, and coupled to an FID. A fused silica capillary column, SPTM-2560 (100 mm × 0.25 mm i.d., 0.2 μm film thickness, Agilent), was used. GC-FID analysis conditions included an injector temperature of 250 °C, injection in splitless mode, and a gas flow rate of 0.8 mL/min. The oven temperature was programmed as follows: initial temperature 100 °C (held for 4 min), increased at 6 °C/min to 160 °C, and then increased at 3.5 °C/min to 240 °C (held for 13 min).

#### 2.3.5. Analysis of Sterols

The determination of individual sterols, total sterols, and triterpene dialcohols (erythrodiol and uvaol) was performed according to the latest version of the IOC method COI/T.20/Doc.26. A Shimadzu (GC-2010) gas chromatograph equipped with a flame ionization detector (FID), a DB-5 (30 m × 0.32 mm × 0.25 μm) capillary column, and an auto-sampler injector was used for the GC analysis (Nishinokyo Kuwabara-cho, Nakagyo-ku, Kyoto 604-8511, Japan). Cholesterol, 24-methylen-cholesterol, campesterol, campestanol, stigmasterol, Δ7-campesterol, Δ5,23-stigmastadienol, clerosterol, β-sitosterol, sitostanol, Δ5-avenasterol, Δ5,24-stigmastadienol, Δ7-stigmastenol, Δ7-avenasterol, erythrodiol, and uvaol (calculated as total erythrodiol) were quantified using α-cholestanol as the internal standard. Results were expressed as proportions (%) of total sterols and total sterols expressed as mg/kg.

### 2.4. Statistical Analysis

Statistical analyses were performed using XLSTAT software (version 2023.2.1413) to evaluate the inheritance of agronomic traits and olive oil parameters in 32 genotypes from the reciprocal crosses ‘Sikitita’ × ‘Arbosana’ and ‘Arbosana’ × ‘Sikitita’. Three primary statistical techniques were employed: descriptive analysis (Scattergram plots), analysis of variance (ANOVA), and principal component analysis (PCA) [54].

#### 2.4.1. Analysis of Variance (ANOVA)

ANOVA was conducted to determine the statistical significance of differences among the means of the agronomic traits and olive oil parameters. This analysis helps to identify whether the observed variations among different genotypes (or groups of genotypes) are due to genetic differences or random variability. The following steps were followed for the ANOVA:Data Preparation: The dataset was organized with the genotypes (and/or reciprocal cross groups) as the independent variable and the agronomic traits and olive oil parameters as dependent variables. Logarithmic transformation was implemented to normalize the data.ANOVA Execution: A one-way ANOVA was conducted for each trait and parameter to test the null hypothesis that there are no significant differences among the genotype means.post hoc Testing: When significant differences were detected, post hoc tests (e.g., Tukey’s HSD) were performed to identify specific pairs of genotypes with significant differences.

#### 2.4.2. Principal Component Analysis (PCA)

PCA was utilized to reduce the dimensionality of the dataset and to identify the principal components that explain the most variance in the data. This technique helps in simplifying the complexity of the data while retaining the most important information. The PCA was carried out as follows:Standardization: The dataset was standardized to ensure that each trait and parameter contributed equally to the analysis.Computation of Principal Components: The eigenvalues and eigenvectors of the covariance matrix were calculated to derive the principal components.Interpretation of Results: The principal components were examined to understand the proportion of variance explained by each component. Biplots were generated to visualize the relationships between the genotypes and the principal components.Analysis of Loadings: The loadings of each variable on the principal components were analyzed to identify the traits and parameters that contributed most to each component.

These statistical analyses collectively provided a comprehensive understanding of the inheritance patterns of agronomic traits and olive oil parameters, facilitating the identification of key genetic influences and their potential applications in olive breeding programs.

## 3. Results

### 3.1. Agronomic Characteristics

The agronomic characteristics analyzed did not show significant differences between the progenies of the reciprocal crosses or the parental varieties ‘Arbosana’ and ‘Sikitita’, as illustrated in (Figure 1). However, significant differences in tree width were observed with the reference cultivar ‘Arbequina’, which was not used as a parent.

Specifically, the ANOVA for eleven agronomic traits in the progenies resulting from the cross ‘Sikitita’ × ‘Arbosana’ and its reciprocal ‘Arbosana’ × ‘Sikitita’ revealed no significant differences (*p* > 0.05) between the reciprocal crosses in all traits, including tree height, tree width, trunk diameter, flesh/stone ratio, dry matter oil content, fruit moisture, ripening index, average fruit weight, average endocarp weight, and oxidative stability, as shown in Figure 1 and Table 1.

The lack of significant differences between groups can be attributed to the high variability among the progenitors of the same cross, leading to high standard errors that make it difficult to identify significant differences between reciprocal crosses. However, descriptive statistical analyses, such as mean and median, show a slight maternal effect tendency in reciprocal crosses.

The mean fruit ripening index (RI) values showed slight, non-significant differences among the reciprocal crosses ‘Arbosana’ × ‘Sikitita’ (1.3) and ‘Sikitita’ × ‘Arbosana’ (1.7) (Appendix A). ‘Sikitita’ is characterized by an early maturation and higher RI compared to ‘Arbosana’. Similar results were observed for tree height and width, with mean values of 312 cm and 270 cm, respectively, for the ‘Arbosana’ × ‘Sikitita’ cross, and 302 cm and 258 cm, respectively, for the ‘Sikitita’ × ‘Arbosana’ cross.

A similar pattern is observed in most agronomic traits, where, despite statistically non-significant differences due to high variability in expression among progenitors of the same cross (Figure 2), descriptive analyses (mean and median) tend to indicate a maternal effect in the main agronomic traits, especially when they are well differentiated between the maternal and paternal parents (Appendix A).

### 3.2. Elaiotechnical Characterization

#### 3.2.1. Fatty Acids

Similarly to the agronomic traits, the olive oil chemical properties were analyzed to further examine a potential maternal effect. The analysis of the fatty acid profile in extra virgin olive oil (VOO) obtained from progenies of the reciprocal crosses ‘Arbosana’ × ‘Sikitita’ and ‘Sikitita’ × ‘Arbosana’ showed no significant differences in the variables saturated fatty acids (SFAs) (R² = 0.020, *p* > 0.05) and monounsaturated fatty acid (MUFAs) (R² = 0.089, *p* > 0.05), and a slightly significant difference in polyunsaturated fatty acids (PUFAs) (R² = 0.124, *p* < 0.05) (Figure 3A and Table 1). The last is explained by the high level of PUFAs in ‘Arbequina’ (14.6%) compared to their value in the rest of the cultivars and progenies analyzed: 8.1, 9.6, and 8.2% exhibited in ‘Arbosana’, ‘Sikitita’ × ‘Arbosana’, and ‘Arbosana’ × ‘Sikitita’, respectively. The mean values of SFAs were similar between the reciprocal crosses, with averages of 17.6 in the progeny of ‘Arbosana’ × ‘Sikitita’ and 18.1 in the progeny of ‘Sikitita’ × ‘Arbosana’. Likewise, the mean values of PUFAs were 8.2% in the progeny of ‘Arbosana’ × ‘Sikitita’ and 9.6% in the progeny of ‘Sikitita’ × ‘Arbosana’. The MUFA content exhibited a slightly higher mean value in the progeny ‘Arbosana’ × ‘Sikitita’ than in the progeny of ‘Sikitita’ × ‘Arbosana’, with a mean concentration of 74.1% and 72.2%, respectively (Appendix A).

The control cultivar ‘Arbequina’, which was not used as a parent, showed significant differences with both reciprocal progenies and ‘Arbosana’, while no significant differences were observed between the reciprocal progenies and the ‘Arbosana’ parent. This result demonstrates the high transmission ratio of traits between parents and progenies.

To further explore the maternal effect on the fatty acid profile, a PCA was performed to represent the distribution of the analyzed samples (fatty acid profile—SFAs, MUFAs, and PUFAs—of each analyzed tree) in a Biplot graph (Figure 4). The distribution of the point clouds shows that the samples from vegetatively reproduced trees of the two traditional cultivars ‘Arbequina’ and ‘Arbosana’ tend to be less variable, as the points of each cultivar are closely grouped, while the samples from sexually reproduced progenitors (seeds) of the reciprocal crosses exhibit wide variance in fatty acids, with dispersed point clouds. Additionally, most samples from the ‘Arbosana’ × ‘Sikitita’ crossing are concentrated towards the ‘Arbosana’ (maternal parent) point cloud and, like the maternal parent, tend to be richer in PUFAs compared to the reciprocal progenitors ‘Sikitita’ × ‘Arbosana’ (Figure 4 and Figure 5).

#### 3.2.2. Phenolic Compounds

The analysis of the phenolic profile showed no statistically significant differences between progenies of the reciprocal crosses tested in terms of total phenols (*R*² = 0.042, *p* > 0.05). However, the findings revealed substantial variation in terms of mean and median for the total phenol concentration among progenies derived from reciprocal crosses. Specifically, the total phenolic content ranged from 13 to 650 mg/kg in progenies from the ‘Arbosana’ × ‘Sikitita’ crossing, and from 27 to 629 mg/kg in progenies from the ‘Sikitita’ × ‘Arbosana’ reciprocal crossing (Figure 6A,B). The most abundant phenolic compounds were Oleacein, Oleocanthal, Oleuropein Aglycon, and Ligustroside Aglycon. These results are in line with the results observed in previous studies [12,55,56,57].

The median values suggest that the progenies coming from the ‘Sikitita’ × ‘Arbosana’ crossing generally exhibit a higher phenolic concentration, with a median total phenol content of 203 mg/kg, compared to 180 mg/kg in the set from the ‘Arbosana’ × ‘Sikitita’ reciprocal cross (Figure 3C), (Appendix A). This observation may be attributed to a slight maternal effect of the ‘Sikitita’ variety which in other studies has shown a higher phenolic content (up to 20%) compared to the ‘Arbosana’ cultivar [58] (Figure 6A–C). Also, as shown in the same figures, the phenolic content of the reference cultivars of the super-intensive planting system (‘Arbequina’ and ‘Arbosana’) is usually relatively low, around 200–300 mg/kg, while the reference cultivars, which are highly phenolic-rich, exceed 1000 mg/kg [58,59]. On the other hand, it is observed that the genitors obtained in the reciprocal crosses show a high variability among them, and a fraction of them (approximately 20% of the seedlings in the case of the cross ‘Sikitita’ × ‘Arbosana’) show a high phenolic concentration (between 500 and 700 mg/kg) compared to the parents and the reference cultivar ‘Arbosana’. This result provides evidence for the possibility of improving the nutritional characteristics of the new cultivars used in the super-intensive system, thus positively influencing the society’s health and biodiversity in olive growing.

#### 3.2.3. Sterols

After analyzing the VOO samples of the tested individuals, a total of 15 sterol compounds were identified, with approximately 96% of their content comprised of β-sitosterol, Δ5-avenasterol, and campesterol. Among these, β-sitosterol emerged as the most prevalent compound, constituting about 85% of the total sterol content (Appendix A).

The analysis of variance revealed no significant differences in the sterol profile between the reciprocal crosses or between ‘Arbosana’ and ‘Arbequina’ (Figure 3B). The variance explained by the maternal effect (reciprocal cross) was minimal and not significant, as observed in the cases of campesterol (R² = 0.046, *p* > 0.05), β-sitosterol (R² = 0.169, *p* > 0.05), and Δ5-avenasterol (R² = 0.140, *p* > 0.05). The sterol content across the various individuals studied was generally similar, except for Δ5-avenasterol. The progenies from the cross between ‘Arbosana’ × ‘Sikitita’ showed a lower Δ5-avenasterol content (5.4%) compared to the rest of the individuals, with values of 7.1, 8.5, and 7.6% in the ‘Sikitita’ × ‘Arbosana’, ‘Arbosana’, and ‘Arbequina’, respectively.

Furthermore, scattergrams illustrated minimal variability in the sterol profile within the same variety and between genotypes, with β-sitosterol content ranging between 82% and 90%, campesterol content ranging from 2% to 4%, and Δ5-avenasterol content ranging from 2% to 12% (Figure 7). These findings indicate that the sterol profile is stable and exhibits limited variability among genotypes, suggesting an absence of parental effect on the sterol profile of the analyzed reciprocal crosses.

## 4. Discussion

The results of our study revealed significant variability in multiple traits among the various individuals studied. This variability was observed between different progenies as well as within the same trait across different varieties according to previous works [14,32,60]. The highest coefficients of variation (CVs) were observed in the ripening index for agronomic traits, Hydroxytyrosol for phenolic compounds, PUFAs for fatty acids, and Δ5-avenasterol for sterols. Notably, phenolic content exhibited the widest range of variability across crosses and varieties (Appendix A), aligning with the results found by other authors [56,61,62,63], since these compounds are of great interest in breeding programs due to their essential role in the nutritional properties of oil and commercial properties such as flavor and shelf life. Previous studies have documented comparable ranges of fatty acid content variability [64,65], consistent with our findings. The genetic variability observed in the progenies of reciprocal crosses, as expected, is wider than the variability found within individuals of clonal cultivars, such as ‘Arbequina’ and ‘Arbosana’ (Appendix A).

### 4.1. Parental Effect on Trait Transmission

Based on our investigation into the parental effect on the transmission of traits in olive breeding, we can ascertain that there is no statistically significant maternal influence on the transmission of the totality of the traits examined (α = 0.95). Similar results were obtained by a study of the parental effect on olive floral quality performed by Rapoport et al. (2022) [33] and an investigation conducted by Valverde et al. (2021) [60] studying the maternal effect in resistance to *Verticillium*. The same author also dismissed the existence of any maternal effects on the olive oil content of the reciprocal crosses between ‘Arbosana’ and ‘Sikitita’ [32].

Despite the lack of statistically significant differences between the reciprocal crosses, there is a tendency of a maternal effect in practically all the traits analyzed in terms of descriptive statistics analysis. This trend is evident in the mean and median values of the progeny sets of each reciprocal cross and the PCA and box plot distribution analyses (Figure 2, Figure 4, Figure 5, Figure 6 and Figure 7; Appendix A). The results from the descriptive statistical analysis suggest a slight positive maternal effect on the transmission of traits to offspring. For example, the mean values of MUFA content exhibited a slightly higher mean value in the progeny ‘Arbosana’ × ‘Sikitita’ than in the progeny of ‘Sikitita’ × ‘Arbosana’, with a mean concentration of 74.1% and 72.2%, respectively. Previous studies have shown that ‘Arbosana’ is richer in MUFAs compared to ‘Sikitita’ [65], indicating a slight and non-significant maternal positive effect on the transmission of fatty acid profile traits [51].

However, this effect seems to be minimal. To definitively confirm or rule out the potential maternal effect, future studies with an experimental design featuring a large number of blocks and replicates are necessary.

### 4.2. Mechanisms of Phenotypic Differences and Implications for Breeding Programs

The slight maternal effect observed using descriptive statistical analysis in olive could be explained by the different genetic and epigenetic mechanisms observed in other species.

Accordingly, different mechanisms can lead to phenotypic differences between reciprocal hybrids including maternal effects [66,67], as well as cytoplasmic inheritance [68,69], or genomic imprinting [70,71,72]. Cytoplasmic inheritance results from the unequal distribution of plastids and mitochondria during microgametogenesis or fertilization. Around 80% of angiosperms, such as olives, display a maternal inheritance of organelles [73], with this form of inheritance being specifically observed in tomatoes [74].

This research contributes to olive breeding programs which primarily aim to develop new genotypes suitable for high-density cultivation. Such cultivation requires critical traits, notably, trees of low vigor coupled with high productivity. The efficiency of crossbreeding programs mainly depends on the choice of progenitors and the knowledge of the transmission of traits that we want to improve. 

Regardless of the traits under examination, broad variability is anticipated, with the inclination towards one parent primarily determined by the distribution probabilities of parental genes within the progeny’s genome. Nonetheless, global results and descriptive statistical analyses suggest using the primary progenitor as the maternal parent to ensure the maximum transmission of desired traits to the progeny and to maximize the process known as heterosis [75].

Although, consistently with previous studies, no significant statistical differences were found in the traits examined, it is confirmed that both female and male parents contribute equally to the genetic makeup of the progeny. This finding indicates that in olive breeding programs, either parent can be used as the maternal or paternal contributor, offering flexibility for breeders. This flexibility is particularly advantageous when selecting parents based on factors like stored pollen availability, the number of mother trees, or seasonal variations in tree productivity.

However, the slight and non-significant maternal influence on characters transmission observed in the descriptive statistical analysis suggests that further investigation is warranted. Experimental trials increasing the number of parents could provide deeper insights into this inheritance of characteristic from parents to seedling.

## Figures and Tables

**Figure 1 plants-13-02467-f001:**
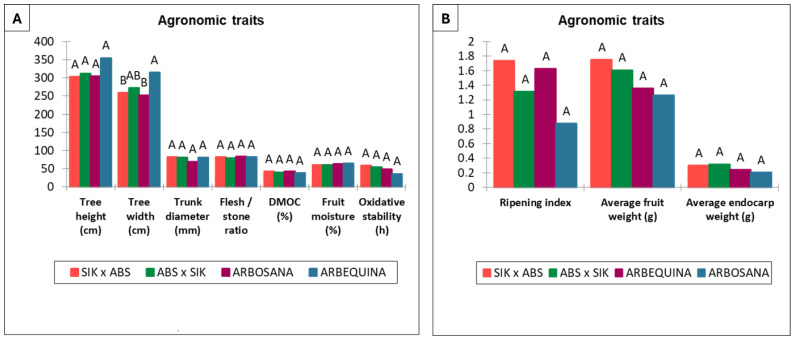
Mean values and statistical differences found through ANOVA analysis for the agronomic traits studied in reciprocal crosses and the reference cultivars ‘Arbosana’ and ‘Arbequina’. Means with the same letter are not significantly different according to the High Significant Difference Test at *p* ≤ 0.05. (SIK—‘Sikitita’, ABS—‘Arbosana’). (**A**) Parameters include tree height, tree width, trunk diameter, flesh/stone ratio, DMOC (dry matter oil content), fruit moisture, and oxidative stability. (**B**) Parameters include ripening index, average fruit weight, and average endocarp weight.

**Figure 2 plants-13-02467-f002:**
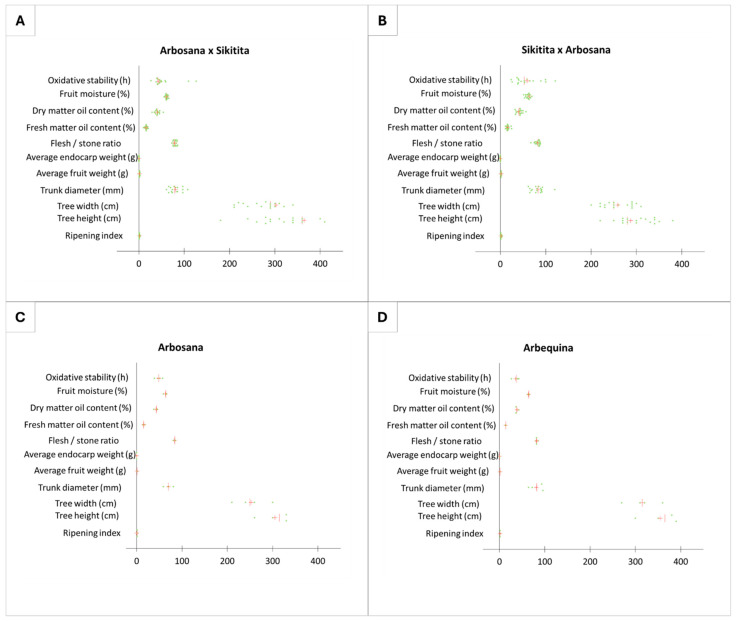
Agronomic characters variability represented in the dot plot graph from the reciprocal cross ‘Arbosana’ × ‘Sikitita’ (**A**), ‘Sikitita’ × ‘Arbosana’ (**B**), the reference cultivar ‘Arbosana’ (**C**), and the reference cultivar ‘Arbequina’ (**D**). Each dot represents a sample analyzed for a given trait, the red vertical line represents the median, and the red cross represents the mean.

**Figure 3 plants-13-02467-f003:**
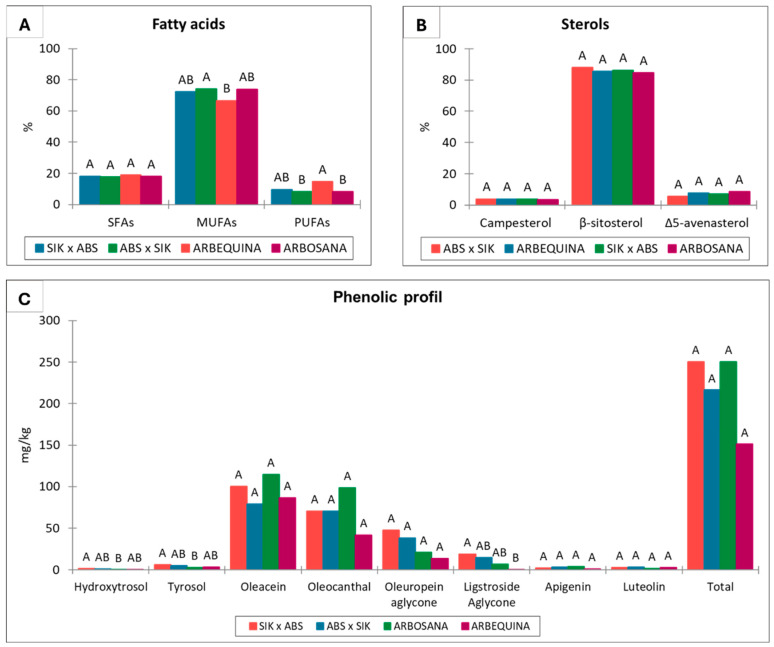
Mean values and statistical differences found using ANOVA analysis for the oil quality parameters studied in reciprocal crosses and the reference cultivars ‘Arbosana’ and ‘Arbequina’. Means with the same letter are not significantly different according to the High Significant Difference Test at *p* ≤ 0.05. (SIK—‘Sikitita’, ABS—‘Arbosana’). (**A**) Fatty acid profile, (**B**) sterol profile, and (**C**) phenolic profile. (SFAs: saturated fatty acids; MUFAs: monounsaturated fatty acids; PUFAs: polyunsaturated fatty acids).

**Figure 4 plants-13-02467-f004:**
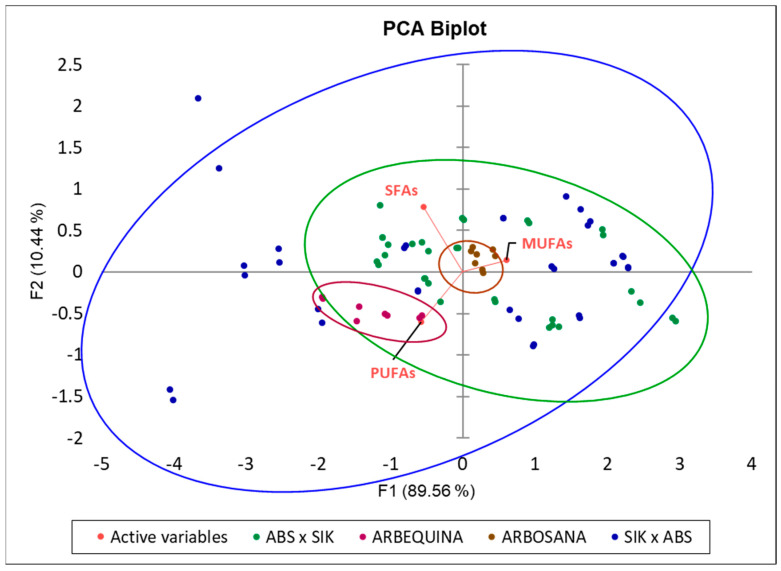
Principal component analysis (PCA) displaying the fatty acid profile variation (loadings: SFAs: saturated fatty acids; MUFAs: monounsaturated fatty acids; PUFAs: polyunsaturated fatty acids) in all the samples analyzed (scores represented by dots). Each cultivar and each reciprocal cross (SIK—‘Sikitita’; ABS—‘Arbosana’) are represented by a specific color. The ellipses represent the distribution and tendencies of the samples and the clustering of each genotype or reciprocal cross group of samples.

**Figure 5 plants-13-02467-f005:**
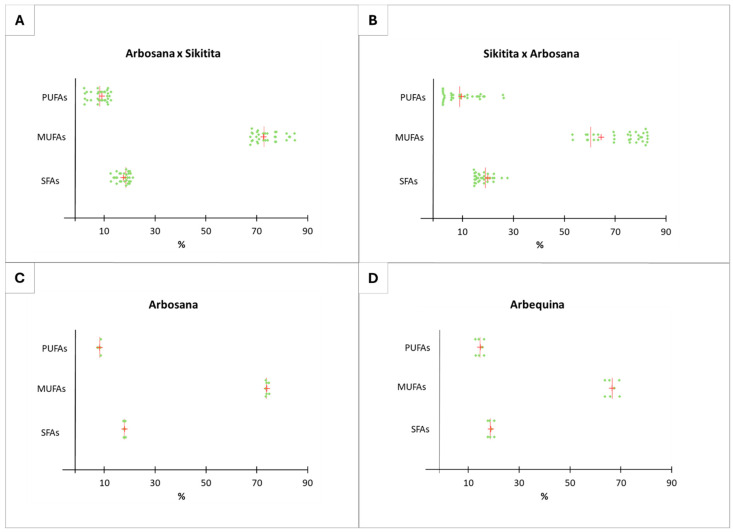
Olive oil fatty acid variability represented in the dot plot graph from the reciprocal cross ‘Arbosana’ × ‘Sikitita’ (**A**), ‘Sikitita’ × ‘Arbosana’ (**B**), the reference cultivar ‘Arbosana’ (**C**), and the reference cultivar ‘Arbequina’ (**D**). Each dot represents a sample analyzed for a given trait, the red vertical line represents the median, and the red cross represents the mean. (SFAs: saturated fatty acids; MUFAs: monounsaturated fatty acids; PUFAs: polyunsaturated fatty acids).

**Figure 6 plants-13-02467-f006:**
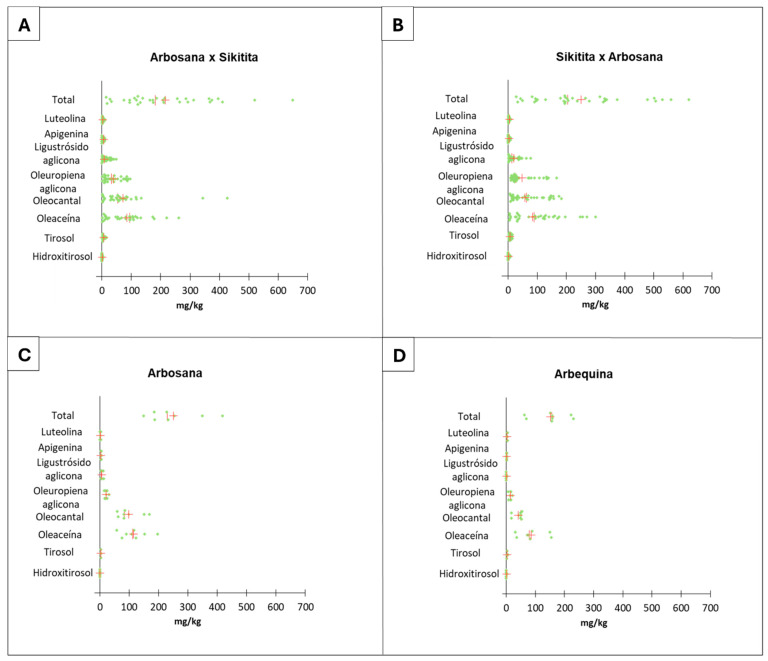
Olive oil phenolic profile variability represented in the dot plot graph from the reciprocal cross ‘Arbosana’ × ‘Sikitita’ (**A**), ‘Sikitita’ × ‘Arbosana’ (**B**), the reference cultivar ‘Arbosana’ (**C**), and the reference cultivar ‘Arbequina’ (**D**). Each dot represents a sample analyzed for a given trait, the red vertical line represents the median, and the red cross represents the mean.

**Figure 7 plants-13-02467-f007:**
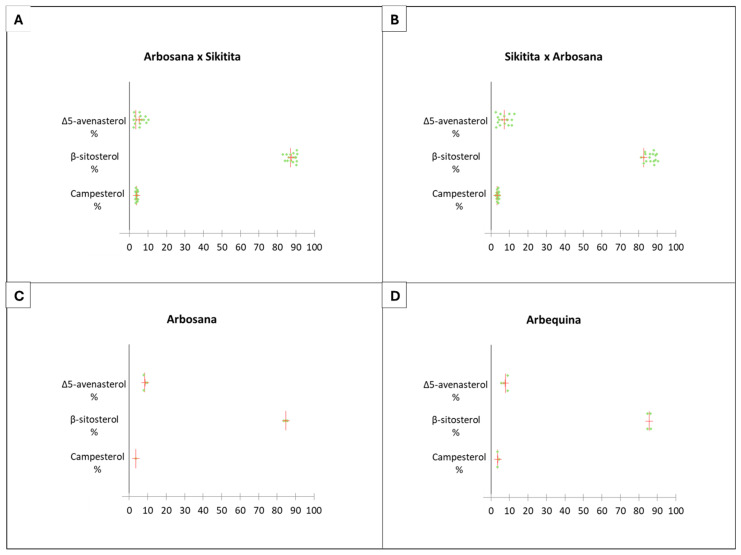
Olive oil sterol profile variability represented in the dot plot graph from the reciprocal cross ‘Arbosana’ × ‘Sikitita’ (**A**), ‘Sikitita’ × ‘Arbosana’ (**B**), the reference cultivar ‘Arbosana’ (**C**), and the reference cultivar ‘Arbequina’ (**D**). Each dot represents a sample analyzed for a given trait, the red vertical line represents the median, and the red cross represents the mean.

**Table 1 plants-13-02467-t001:** This table presents the ANOVA results, detailing the impact of the reciprocal cross factor on the variance of olive agronomic traits, phenolic compounds, fatty acids, and sterol profiles in virgin olive oil. Included are the R² coefficient (proportion of variability explained by the studied factor), F-value, *p*-value, and type III sum of squares.

Agronomic Traits											
Parameters	Ripening Index	Tree Height	Tree Width	Trunk Diameter	Average Fruit Weight	Average Endocarp Weight	Flesh/Stone Ratio	Fresh Matter Oil Content	Dry Matter Oil Content	Fruit Moisture	Oxidative Stability
*R* ^2^	0.13	0.09	0.19	0.07	0.11	0.19	0.09	0.08	0.06	0.09	0.07
F	1.88	1.26	2.84	0.88	1.54	2.83	1.29	1.01	0.78	1.26	0.97
*p*-value	0.15	0.30	0.05	0.46	0.22	0.05	0.29	0.40	0.51	0.30	0.42
*Type III SS*	3.10	8990.96	11,509.51	493.79	1.07	0.05	79.23	27.42	77.99	52.65	1853.21
Phenolic compounds											
Parameters	Hydroxytyrosol	Tyrosol	Oleacein	Oleocanthal		Oleuropein aglycone	Ligstroside Aglycone		Apigenin	Luteolin	Total
*R* ^2^	0.15	0.13	0.03	0.03		0.09	0.13		0.12	0.04	0.04
*F*	4.42	3.93	0.78	0.89		2.70	3.77		3.41	1.20	1.14
*p*-value	0.01	0.01	0.51	0.45		0.05	0.01		0.02	0.32	0.34
*Type III SS*	29.21	118.07	11,905.23	12,891.76		9923.75	2641.87		58.54	17.42	73,047.12
Fatty acids						Sterols					
Parameters	SFA		MUFA	PUFA		Campesterol			β-sitosterol		Δ5-avenasterol
*R* ^2^	0.02		0.09	0.12		0.05			0.17		0.14
*F*	0.52		2.54	3.69		0.60			2.51		2.00
*p*-value	0.67		0.06	0.02		0.62			0.07		0.13
*Type III SS*	11.22		385.93	274.39		0.44			45.74		44.47

## Data Availability

Data can be acquired upon request from the corresponding author.

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
