# Peer review of "Parental Effect on Agronomic and Olive Oil Traits in Olive Progenies from Reciprocal Crosses"

_plants, 2024, doi:10.3390/plants13172467_

Round 1
Reviewer 1 Report
Comments and Suggestions for Authors
Dear Authors,
Suggestions for corrections are in the attached file.
Best regards,

Comments on the Quality of English LanguageDear Authors,
Suggestions for corrections are in the attached file.
Best regards,
Author Response
- Introduction
L32.
The introduction could be enriched with a more comprehensive literature review on the
topic. I suggest including more recent studies that address technological and methodological
advances in the research area. This will give the reader a more up-to-date and complete view of the state of the art.
Thank you for the suggestion. We have added and rephrased some paragraphs in the introduction and also, we have added new updated literature, such as:
“Long-term secondary prevention of cardiovascular disease with a Mediterranean diet and a low-fat diet (CORDIOPREV): a randomised controlled trial
New precision-breeding law unlocks gene editing in England Caccamo MNature Biotechnology
The global need for plant breeding innovation Jorasch PTransgenic Research (2019) 28”
L117. It would be useful to better specify the study's objectives, highlighting the specific gap
that the article seeks to fill. The current introduction addresses the topic generally, but a more
precise delimitation of the research problem would help justify the study's relevance.
Thank you for the suggestion. We have added the next paragraph for improving the objectives definition: “To address the knowledge gap concerning the inheritance of traits related to olive agronomic traits and especially oil quality, this study aims to elucidate how specific characteristics are inherited from parent plants to seedlings. The objectives of the study were to investigate the influence of female and male parents on the inheritance of key agronomic traits, including tree vigor, fruit weight, stone weight, and fat yield, as well as crucial chemical characteristics of olive oil, such as the phenolic profile, stability, fatty acid composition, and sterol content. To achieve that, we conducted a comprehensive evaluation of 32 genotypes derived from reciprocal crosses between the olive cultivars 'Sikitita' and 'Arbosana.' By comparing the outcomes of the 'Sikitita' × 'Arbosana' and 'Arbosana' × 'Sikitita' crosses, this research seeks to provide valuable insights into the inheritance patterns of these traits, thereby contributing to the optimization of olive breeding programs and cultivation practices.”
- MATERIAL AND METHODS.
L119. The description of the methods could be detailed more clearly. For example, the
technique used for data collection could be better explained by indicating the controlled variables
and specific experimental conditions. This would increase the reproducibility of the study.
Correction 2: Considering the importance of methodological precision, I suggest including data on
the calibration of the instruments used, as well as the justification for the choice of statistical
methods employed. This inclusion would strengthen the validity of the results.
According to your comments, the materials and methods section has been improved by better describing the methodology for better reproduction of the experiment.
- RESULTS.
L244. The presented results could be supplemented with additional graphs or tables that facilitate the visualization of the observed trends. Furthermore, it would be beneficial to discuss the results in light of comparative studies, highlighting agreements and disagreements with the existing literature.
Thank you for the suggestion. Due to the large amount of data, we have included all of them in Supplementary Table 1. This ensures that the reader has all the data accompanying the graphs in the main text. We have improved the discussion section.
L312. Phenolic compounds.
I suggest a more in-depth discussion of the practical implications of the results for the agricultural sector. This includes the possible application of the findings in sustainable agricultural practices or public policies, which would increase the study's relevance to the journal's target audience.
Thank you for the suggestion. We have further explained and commented the results related to the phenols content and the possibilities to improve olive oil quality in the new breeding programs (red text).
L561. It would be interesting to include a brief section on the study's limitations, which would provide a more balanced view of the findings and help contextualize the conclusions within a broader framework of scientific uncertainties.
Thank you for your comment. The text has been adapted for better contextualization. Regarding the limitations section, the authors have agreed not to include it. For the authors, one of the “limitations” is not having planted the Sikitita variety when the experiment was planted, although this variety does not influence the results that are being studied in this article since the important thing is the comparison between reciprocal crosses.
L582. The conclusion could be more emphatic regarding the innovative contributions of the study. I recommend highlighting how the results advance knowledge in the area and suggesting specific directions for future research.
We have improved the conclusions highlighting the innovative contributions and study limitation and work that should be continued adding the following text:
“Although, consistent with previous studies, no significant statistical differences were found in the traits examined, it is confirmed that both female and male parents con-tribute equally to the genetic makeup of the progeny. This finding indicates that in olive breeding programs, either parent can be used as the maternal or paternal contribute-tor, offering flexibility for breeders. This flexibility is particularly advantageous when selecting parents based on factors like stored pollen availability, the number of mother trees, or seasonal variations in tree productivity.
However, the slight and non-significant maternal influence on characters transmission observed in the descriptive statistical analysis suggests that further investigation is warranted. Larger experimental trials, with increased replication and balanced parental representation, could provide deeper insights into this potential maternal effect.”

Reviewer 2 Report
Comments and Suggestions for Authors
The presented manuscript: "Parental Effect on Agronomic and Olive Oil Traits in Olive Progenies from Reciprocal Crosses" evaluate the parental effect on different agronomic and olive oil characteristics and its role in breeding programs.
The following changes are recommended and some clarifications should be made:
Introduction
Line 40: Olea europaea should be italic. Therefore, it can be used as O. europea throughout the manuscript.
Line 115-117: The aim of the study should be clearly presented. Also, the novelty of the study could be emphasized at the end of Introduction section.
Materials and Methods
Line 123: The authors mentioned that the trial includes ‘Arbequina’ and ‘Arbosana’ varieties, but thereafter they used ‘Arbosana’ × ‘Sikitita’ and ‘Sikitita’ × ‘Arbosana’ crossings. This should be better explained.
Line 154: The authors should describe how the phenolic compounds were identified and quantified in the samples.
Line 166: What is the measurement unit for oxidation stability index?
Figure 1: It is very cumbersome to present all agronomic traits in Figure a and b. Maybe, it is better to use Table for these data. Why the authors did not present the error bars for all data in Figure 1?
Results
This paragraph should be included in Discussion section: “Previous studies have shown that ‘Arbosana’ is richer in MUFAs compared to ‘Sikitita’ [45], indicating a slight and non-significant maternal effect on the transmission of fatty acid profile traits [41].” Please, check this for other similar discussion statements (Line 321).
Figure 3: All abbreviations in Figures should be defined (for example SFA, MUFA, PUFA). This should be implemented for all Figures. Also, the measurement units for phenolics and fatty acids are not presented.
Why the authors presented PCA plot only for fatty acids, but not for phenolic compounds, sterols and agronomic traits?
Discussion
Discussion section needs significant improvement. The authors should discuss presented results and explain the variation between crossings and progenitors related to the agronomic traits, fatty acid, sterol and phenolic profile.
Is there any conclusion from outgoing results or potential application of the study?
Author Response
Comments and Suggestions for Authors R2
The presented manuscript: "Parental Effect on Agronomic and Olive Oil Traits in Olive Progenies from Reciprocal Crosses" evaluate the parental effect on different agronomic and olive oil characteristics and its role in breeding programs.
The following changes are recommended and some clarifications should be made:
Introduction
Line 40: Olea europaea should be italic. Therefore, it can be used as O. europea throughout the manuscript.
Thank you for the observation. It has been changed.
Line 115-117: The aim of the study should be clearly presented. Also, the novelty of the study could be emphasized at the end of Introduction section.
We have clarified the aim of the study. The next paragraph has been added: “To address the knowledge gap concerning the inheritance of traits related to olive agronomic traits and especially oil quality, this study aims to elucidate how specific characteristics are inherited from parent plants to seedlings. The objectives of the study were to investigate the influence of female and male parents on the inheritance of key agronomic traits, including tree vigor, fruit weight, stone weight, and fat yield, as well as crucial chemical characteristics of olive oil, such as the phenolic profile, stability, fatty acid composition, and sterol content. To achieve that, we conducted a compre-hensive evaluation of 32 genotypes derived from reciprocal crosses between the olive cultivars 'Sikitita' and 'Arbosana.' By comparing the outcomes of the 'Sikitita' × 'Ar-bosana' and 'Arbosana' × 'Sikitita' crosses, this research seeks to provide valuable in-sights into the inheritance patterns of these traits, thereby contributing to the optimi-zation of olive breeding programs and cultivation practices.”
Materials and Methods
Line 123: The authors mentioned that the trial includes ‘Arbequina’ and ‘Arbosana’ varieties, but thereafter they used ‘Arbosana’ × ‘Sikitita’ and ‘Sikitita’ × ‘Arbosana’ crossings. This should be better explained.
The Arbosana and Arbequina varieties are currently the main varieties in hedgerow olive groves. They currently represent around 80% of hedgerow olive groves. They are also the best characterized varieties, so they were used as control/reference varieties for agronomic characteristics. Currently, and after the experiment, the authors have observed that planting the Sikitita variety would have been interesting, although it would not have provided knowledge of how the characteristics are inherited in the offspring, since this is provided to us by the genotypes of the reciprocal crosses included in the experiment.
Line 154: The authors should describe how the phenolic compounds were identified and quantified in the samples.
According to your comments, the materials and methods section has been improved by better describing the methodology for better reproduction of the experiment.
Line 166: What is the measurement unit for oxidation stability index?
The measurement unit is “hours”. We have rephrased the paragraph to let it clear: “In this analysis, 3.2 g of each VOO sample was heated to 100 °C with an air flow rate of 10 L/h to determine the time (in hours) until oil oxidation occurred.”
Figure 1: It is very cumbersome to present all agronomic traits in Figure a and b. Maybe, it is better to use Table for these data. Why the authors did not present the error bars for all data in Figure 1?
Thank you for your observation. All results shown graphically in figure 1 are also represented numerically in the ‘Supplementary Table 1’. There the standard error and all relevant statistical parameters are described. In figure 1 no error bars (but letters showing significant differences/non-significant differences) have been used for aesthetic reasons and because the standard error that the error bars would represent is described in ‘Supplementary Table 1’.
Results
This paragraph should be included in Discussion section: “Previous studies have shown that ‘Arbosana’ is richer in MUFAs compared to ‘Sikitita’ [45], indicating a slight and non-significant maternal effect on the transmission of fatty acid profile traits [41].” Please, check this for other similar discussion statements (Line 321).
The paragraph has been moved and improved the discussion section.
Figure 3: All abbreviations in Figures should be defined (for example SFA, MUFA, PUFA). This should be implemented for all Figures. Also, the measurement units for phenolics and fatty acids are not presented.
Thank you for the note. The abbreviations are defined, and the measurement units are added.
Why the authors presented PCA plot only for fatty acids, but not for phenolic compounds, sterols and agronomic traits?
We present only the fatty acids in PCA as an example to visualise the slight maternal effect on fatty acids, knowing that they are an important and stable trait ( they are not affected much by the environment and are mainly controlled by the genetic factor).
The rest of the traits evaluated are compared by using dot plots graphs in a more summarised way. We would not be able to compare all traits through a PCA because of the high number of plots that would be generated.
Discussion
Discussion section needs significant improvement. The authors should discuss presented results and explain the variation between crossings and progenitors related to the agronomic traits, fatty acid, sterol and phenolic profile.
Our primary objective was to compare whether there were differences between the two groups of reciprocal crosses (regardless of the parental lines), and our findings showed no significant differences. This indicates that the parental lines contribute equally to the transmission of traits, a point that has been reinforced and clarified in our discussion section.
However, comparing the results between seedlings and progenitors, particularly in cases where slight differences were observed in the descriptive analyses, emerged as an additional outcome that enriches the article. Since just one of the progenitors were present in the trial, we referenced published data for the progenitors (Arbosana and Sikitita) for the most important traits (such as fatty acids and phenols) to provide context. Based on this strategy, we have improved the wording in the results and discussion sections to reflect these insights more clearly.
Is there any conclusion from outgoing results or potential application of the study?
A last paragraph has been added with the principal conclusion.

Round 2
Reviewer 2 Report
Comments and Suggestions for Authors
The authors responded satisfactorily to all suggestions. Maybe, editorial/technical/grammatical improvement is necessary.